# Diagnosis of Laryngopharyngeal Reflux: Past, Present, and Future—A Mini-Review

**DOI:** 10.3390/diagnostics13091643

**Published:** 2023-05-07

**Authors:** Han-Chung Lien, Ping-Huan Lee, Chen-Chi Wang

**Affiliations:** 1Division of Gastroenterology, Taichung Veterans General Hospital, Taichung 407219, Taiwan; 2School of Medicine, National Yang Ming Chiao Tung University, Taipei 11217, Taiwan; 3Department of Post-Baccalaureate Medicine, College of Medicine, National Chung Hsing University, Taichung 402, Taiwan; 4Department of Otolaryngology, Taichung Veterans General Hospital, Taichung 407219, Taiwan

**Keywords:** hypopharyngeal multichannel intraluminal impedance-pH, laryngopharyngeal reflux, pharyngeal acid reflux episodes

## Abstract

Laryngopharyngeal reflux (LPR) is a variant of gastroesophageal reflux disease (GERD) in which gastric refluxate irritates the lining of the aerodigestive tract and causes troublesome airway symptoms or complications. LPR is a prevalent disease that creates a significant socioeconomic burden due to its negative impact on quality of life, tremendous medical expense, and possible cancer risk. Although treatment modalities are similar between LPR and GERD, the diagnosis of LPR is more challenging than GERD due to its non-specific symptoms/signs. Due to the lack of pathognomonic features of endoscopy, mounting evidence focused on physiological diagnostic testing. Two decades ago, a dual pH probe was considered the gold standard for detecting pharyngeal acidic reflux episodes. Despite an association with LPR, the dual pH was unable to predict the treatment response in clinical practice, presumably due to frequently encountered artifacts. Currently, hypopharygneal multichannel intraluminal impedance-pH catheters incorporating two trans-upper esophageal sphincter impedance sensors enable to differentiate pharyngeal refluxes from swallows. The validation of pharyngeal acid reflux episodes that are relevant to anti-reflux treatment is, therefore, crucial. Given no diagnostic gold standard of LPR, this review article aimed to discuss the evolution of objective diagnostic testing and its predictive role of treatment response.

## 1. Introduction

Laryngopharyngeal reflux (LPR) is characterized by individuals who present with chronic laryngopharyngeal symptoms such as hoarseness, vocal fatigue, excessive throat clearing, globus pharyngeus, cough, postnasal drip as well as laryngoscopic signs such as erythema, edema, ventricular obliteration, postcricoid hyperplasia, and pseudosulcus change [1]. Patients may or may not have typical reflux symptoms and, therefore, may visit an otolaryngologist or a gastroenterologist, presumably depending on their primary symptoms. Various non-reflux etiologies such as voice overuse, infection, allergy, or exposure to environmental irritants may also contribute to similar symptoms and signs. Despite the development of “disease-specific” instruments to measure the disease severity such as the Reflux Symptom Index (RSI) [2] and the Reflux Finding Score [3], the symptoms and signs remain “non-specific”. As a result, reflux itself is just one of a myriad of causes which irritate the lining of aerodigestive tract. LPR is a prevalent disease which was estimated to be 10% of the outpatients in the otolaryngology units [1]. The quality of life of LPR patients is generally poor [4]; however, the management is challenging. Traditionally, using empirical proton pump inhibitors (PPIs) once or twice daily is often a pragmatic therapeutic strategy and those who are refractory to high dose PPIs treatment are recommended to refer for the reflux testing [5]. Such an algorithm was recently challenged by the up-front testing using impedance-pH and manometry prior to anti-reflux therapy in order to minimize the cost [6]. Moreover, there are discrepancies between otolaryngology and gastroenterology guidelines regarding the indications of acid suppression therapy [1,7]. The gastroenterology guidelines recommend against acid suppression therapy in patients with isolated LPR symptoms because there is scarce evidence to show the superiority of PPIs to placebo in controlled trials, while the otolaryngology guideline states that the majority of LPR patients do not have heartburn or esophagitis, i.e., isolated LPR symptoms. Recent Lyon consensus for diagnosis of gastroesophageal reflux disease (GERD) also questioned the utility of proximal esophageal or pharyngeal testing because of the lack of consistent outcome studies [8]. The aim of this review is to discuss the evolution of objective testing for LPR and its predictive role on anti-reflux therapy.

## 2. Definition and Disease Burden of LPR

### 2.1. Definition

Numerous terms describe airway symptoms/signs caused by gastroesophageal reflux, such as reflux laryngitis, laryngeal reflux, gastropharyngeal reflux, pharyngoesophageal reflux, supraesophageal reflux, extraesophageal reflux, or atypical reflux [1]. In 2002, the American Academy of Otolaryngology—Head and Neck Surgery position statement used the term LPR, defined as “the backflow of stomach contents into the throat, that is, into the laryngopharynx”. However, under the conceptual definition of LPR, which mainly focused on the direct contact of refluxate into the lining of upper airway, a subset of patients with a reflexogenic mechanism of symptom generation, i.e., the stimulation of a vagal reflex arc, could be underestimated [9]. This notion was supported by our recent data showing that patients with isolated LPR symptoms (ILPRS) and pathological esophagopharyngeal (either esophageal or pharyngeal) reflux may have much fewer pharyngeal acid reflux episodes than their counterparts who have concomitant typical reflux symptoms (CTRS) [10]; while both distal esophageal acid exposure and response rate to PPIs treatment were similar between the two, suggesting a reflexogenic mechanism in patients with ILPRS [11]. The Montreal definition describes “GERD is a condition which develops when the reflux of gastric content causes troublesome symptoms” and uses the term “extraesophageal syndromes of GERD” for LPR symptoms [12]. It is now clear that heartburn and regurgitation, the cardinal symptoms of GERD, do not equate GERD [13]. This is even more true for LPR. Unfortunately, most clinical trials that adopted the LPR symptom severity with or without laryngeal signs as the only inclusion criteria in this inherently heterogeneous group may have inevitably generated heterogeneity between studies [2,14]. In that sense, we consider LPR or extraesophageal reflux as a variant of GERD in which gastric refluxate irritates the lining of the aerodigestive tract and causes troublesome airway symptoms or complications.

### 2.2. Burden of LPR

Given that there are no specific laryngeal symptoms/signs and no established diagnostic gold standard for LPR, its prevalence is unclear. A survey of general practice in the UK used an LPR-specific questionnaire, the RSI, and estimated a prevalence of 26.5% among 951 participants based on a cut-off of 10 points of the RSI score [15]. Although the data may not represent the genuine prevalence, the economic burden of caring for patients with LPR was four to five times to that of typical GERD in the US, where PPIs were the single greatest contributor to the cost of LPR management [16]; this indicates a substantial medical burden and significant socioeconomic impact. Notably, the efficacy of PPI treatment on LPR varies. Uncontrolled studies showed that 50% to 70% of patients with LPR responded to PPI therapy [17], whereas controlled trials failed to show the superiority of PPIs over placebo in a meta-analysis [18], suggesting the importance of objective diagnostic testing. In patients with suspected LPR, the sensitivity of esophagoscopy is low [19], while the laryngoscopic findings suggestive of reflux were not associated with pathological MII-pH data [20] and were common in normal volunteers [21]. Thus, it is conceivable that the various etiologies in patients with LPR symptoms may have contributed to the mixed results of the response to PPI therapy [22]. Owing to the vulnerability of airway mucosa to pepsin-containing refluxate [23], current pharmacological therapeutic strategies often adopt an empiric high dose (twice daily) and prolonged PPIs (3 to 6 months) use [5]. Such a therapeutic strategy may not only increase medical cost but also carry an increased risk of gut dysbiosis and potential subsequent complications for long-term users [24]. Liquid alginate suspension is another anti-reflux medication that may be effective in relieving LPR symptoms. Instead of acid suppression, it forms a gel raft to serve as a pH neutral barrier at acid pocket in the proximal stomach to reduce reflux. However, the therapeutic role of alginate in LPR is inconsistent between controlled trials. For example, McGlashan et al. showed that liquid alginate suspension is more effective than no treatment for relieving LPR symptoms [25], and Wilkie el al. found that co-prescription of high dose PPIs with alginate did not offer additional benefit when comparing alginate alone [26]. In contrast, Tseng et al. conducted a double-blind, placebo-controlled trial and found no significant difference between alginate and placebo in relieving LPR symptoms and signs [27]. Notably, all of the above trials only adopted LPR symptoms and signs as the inclusion criteria for study populations. Therefore, there is an urgent need for objective diagnostic biomarkers that may predict response to anti-reflux treatment. 

## 3. Diagnostic Challenges of LPR

### 3.1. The Lack of Validated Objective Testing

In patients with typical GERD, heartburn and regurgitation are cardinal symptoms in which the majority of patients may respond to PPIs therapy. Thus, symptom-based empirical PPIs therapy remains the mainstay therapeutic strategy in patients with a low risk of malignancy. On the other hand, endoscopy may provide evidence of reflux such as reflux esophagitis or Barrett’s esophagus, as well as weak barrier of esophagogastric junction for reflux such as hiatal hernia despite its low sensitivity. However, this is not the case in patients with suspected LPR as neither heartburn nor reflux esophagitis is common in the majority of patients [1]. One study reported that among 128 patients with suspected LPR, only 18% had reflux esophagitis and 0.8% had Barrett’s esophagus, whereas 81% had pathological reflux detected by wireless pH monitoring. Notably, the presence of typical reflux symptoms (heartburn or regurgitation) did not predict the presence of pathological reflux in this study [19]. Therefore, reflux monitoring seems to be a prerequisite for demonstrating evidence of reflux. Until now, several objective reflux tests were developed to diagnose LPR, including dual or triple pH (simultaneous pharyngeal and esophageal pH) monitoring, oropharyngeal pH monitoring, multichannel intraluminal impedance-pH (MII-pH), and hypopharyngeal multichannel intraluminal impedance-pH (HMII-pH). Unfortunately, neither methodology nor interpretation of these tests was standardized. More importantly, outcome studies linked to objective tests are scarce [8].

### 3.2. The Lack of Validated Outcome Data Linked to the Testing

From a clinical point of view, exploring factors that predict response to anti-reflux therapy may shed light on disease pathophysiology and be of diagnostic potential. Therefore, objective biomarkers that predict symptom response to anti-reflux therapy are of paramount importance. In a retrospective study to identify the predictors of response to anti-reflux surgery, the response to acid-suppression therapy before surgery was associated with a long-term response to anti-reflux surgery in patients with LPR symptoms [28], indicating a surrogate marker of clinical outcome. Several reflux-monitoring-based parameters linked to acid-suppression therapy were also investigated. Based on dual pH monitoring, Ulualp et al. retrospectively found that pre-treatment documented pharyngeal acid reflux (PAR) episodes were not associated with symptom response to acid-suppression therapy in 39 patients with posterior laryngitis [29]. However, Park et al. conducted a prospective cohort study in 85 patients with suspected LPR and found that the baseline acid exposure time in both proximal and distal esophagi was marginally higher in the PPI responders than the non-responders [30]. In addition, Wang et al. used MII-pH in 92 patients with suspected LPR and found that both increased distal esophageal acid exposure and increased pharyngeal bolus exposure time may predict response to PPI therapy [31]. Taken together, these data suggest that reflux-monitor-based parameters encompassing both distal and proximal reflux may be more sensitive than those only monitoring either proximal or distal reflux in patients with suspected LPR. 

## 4. Evolution of Diagnostic Modalities

### 4.1. Past: Dual pH Probes Era

#### 4.1.1. The Limitations of Dual pH Probes

In 2002, the ENT statement advocated ambulatory 24-h dual pH (simultaneous esophageal and pharyngeal) monitoring as the gold standard for the diagnosis of LPR [1] and epidemiological data also supported a higher prevalence of PAR episodes in patients with suspected LPR than asymptomatic controls [32]. However, neither accepted universal criteria of PAR episodes nor the threshold of PAR episodes relevant to anti-reflux therapy [29] were established in the past two decades [33]. This is probably because of frequent swallow-related artifacts that interfere in the interpretation of dual pH recording [34], as demonstrated by HMII-pH monitoring [35]. Additionally, the location of hypopharyngeal pH sensors, the placement of catheters with either endoscopy or manometry, and the proposed cut-off number of PAR episodes may also contribute to the determination of pathological PAR [36].

#### 4.1.2. The Proposed Criteria of Candidate PAR Episodes

Using dual pH sensors, in 1999, Williams et al. found that that 92% of pharyngeal pH decreases of 1 to 2 units were definite artifacts due to the lack of simultaneous or preceding esophageal acidification. In contrast, 35 out of 45 (77%) pharyngeal pH decreases of greater than 2 units, with a nadir pH of less than 5 within 30 sec, were temporally associated with simultaneous or preceding esophageal acidification [37]. Using triple pH-sensors catheters, we proposed the aforementioned criteria of candidate PAR episodes and found that 17% of 104 consecutive patients with suspected LPR have candidate PAR episodes that have good-to-excellent inter-observer agreement [38].

#### 4.1.3. The Potential Diagnostic Role of Candidate PAR Episodes

The triple pH sensor is an ambulatory 24 h pH catheter incorporating three pH sensors into a bifurcated probe with a single connector and recording box, and it is able to simultaneously detect acid reflux in the hypopharynx, proximal esophagus, and distal esophagus [39]. In their study, ninety percent of normal participants showed no PAR episodes or a single episode over a 24-h period. Based on the proposed mechanisms involving “reflux” and “reflex” for LPR symptom generation [9], we proposed a composite pH parameter incorporating excessive candidate PAR episodes, i.e., ≥2 episodes/24 h and excessive acid exposure time in the distal esophagus using triple pH sensors. We conducted a prospective cohort study to evaluate the predictability of the proposed composite pH parameter at baseline for the response to PPI therapy in 107 patients with suspected LPR, including 65 with CTRS and 42 with ILPRS. Compared to those with a negative composite pH among patients with ILPRS, we found that participants with a positive composite pH at baseline had a 10-fold and an 8-fold likelihood of predicting a response to PPI therapy at 8-week and 12-week time points, respectively. However, the association was not significant among patients with CTRS, despite the existence of a trend toward a higher response rate in patients with a positive composite pH [40]. One possible explanation for the predictability in patients with ILPRS is that pathological reflux is likely the inducer of laryngeal symptoms (high positive predictive value) because of the high specificity nature of pH-metry despite the low pretesting probability of pathological reflux given the absence of esophageal symptoms. In contrast, in those with CTRS, pathological reflux may be either an inducer, a cofactor, or a bystander, because the pretesting probability of pathological reflux is high irrespective of causation. It is possible that reflux is a cofactor in patients with partial response to PPI therapy and their laryngeal symptoms were partially due to non-reflux etiologies such as allergy. It is also possible that reflux is a bystander in those whose laryngeal symptoms are completely refractory to PPI therapy [41] (Figure 1). Another possible explanation for the poor predictability of pathological reflux to the response to PPI therapy in patients with CTRS and pathological reflux is that these patients are more likely to have excessive PAR episodes than their ILPRS counterparts [10]; thus, direct injury from either weakly acidic or non-acidic refluxate to the larynx may not be eliminated despite the use of high dose acid-suppression therapy [42,43,44]. 

### 4.2. Present: Hypopharyngeal Multichannel Intraluminal Impedance-pH (HMII-pH) Era

#### 4.2.1. Current Objective Pharyngeal Reflux Testing

In addition to dual pH or triple pH sensor tests, current objective tests to diagnose extra-esophageal reflux commonly used in the clinical setting include the salivary pepsin test, oropharyngeal pH monitoring, and the HMII-pH. The salivary pepsin test is a noninvasive diagnostic tool that contains two antibodies to human pepsin and can rapidly detect the presence and quantify the concentration of pepsin in saliva. A positive result indicates the presence of refluxate from the stomach to the mouth. Wang et al. found that strong positive results of salivary pepsin test predicts better PPI response in 74 patients with suspected LPR [45]. However, Yadlapati et al. compared the salivary pepsin concentrations between patients with CTRS, ILPRS, and healthy controls and found that the CTRS group but not the ILPRS group had a higher concentration of salivary pepsin than the control group and there was no significant difference in the positive rate among the three groups, indicating a limited diagnostic role in distinguishing patients from healthy participants [46]. Similarly, oropharyngeal pH monitoring is also unable to distinguish patients with LPR from healthy participants based on the percentage time below pH 4.0, 5.0, 5.5, 6.0, or RYAN score [46]. It also did not predict 12-week PPI therapy in a small-scale prospective cohort study [47]. Although the oropharyngeal pH monitoring is originally designed to detect both liquid and aerosolized form pH, the accuracy in detecting reflux remain uncertain because of frequent artifacts arising from swallows and its poor correlation with simultaneous MII-pH monitoring recording [48,49]. 

#### 4.2.2. Detection of Pharyngeal Acid Reflux Episodes

Twenty-four-hour ambulatory MII-pH monitoring is currently considered the gold standard in diagnosing reflux episodes regardless of the form of gas/liquid or the acidity of refluxate in the Lyon consensus. However, its role in diagnosing LPR remain uncertain [8]. A novel configured MII-pH catheter called HMII-pH, which incorporates two trans-upper esophageal sphincter impedance sensors, was designed to track refluxate along the entire esophagus into the hypopharynx [50]. The preliminary data showed that the median number and the 95-percentile number of pharyngeal liquid or mixed gas–liquid reflux episodes in healthy participants for 24 h were 0 and 0 to 3, respectively [50,51,52]. However, the inter-observer reproducibility of manual analysis of pharyngeal reflux episodes was poor, even when performed by experts [51]; this was presumably due to frequent artifacts encountered from air trapped in between catheters and mucosa, as well as the labor–intensive and time-consuming nature of manual analysis [53]. To reduce the burden of interpretation of non-acid reflux episodes which are often overestimated by automated analysis and are less relevant to acid–suppression therapy, we recently used HMII-pH catheters to evaluate the aforementioned criteria of candidate PAR episodes and found that 80% of 105 candidate PAR episodes were HMII-pH-proven PAR episodes, with an interobserver reproducibility of more than 95% [54] (Figure 2). We also developed a deep-learning-based artificial intelligence model to identify PAR episodes and found a sensitivity of 1.000 and a specificity of 0.909 in the test dataset, indicating the objectivity of the diagnostic criteria of PAR episodes [55]. Future studies should investigate whether the presence of pathological PAR alone is relevant to anti-reflux therapy and symptom–reflux association. 

#### 4.2.3. Prediction of Anti-Reflux Treatment Response Using HMII-pH Parameters

To evaluate the physiological characteristics of patients with ILPRS as well as their response to PPI therapy, we conducted a prospective multi-center study including 398 patients with suspected LPR [10]. A total of 252 patients including 40% PPI-naive patients, underwent either triple pH sensors or HMII-pH catheters when off PPI at baseline. We adopted the aforementioned composite pH criteria and found that 106 patients (42%) had a positive composite pH, including 40 in the ILPRS group and 66 in the CTRS group, and 58% had a negative composite pH. Both ILPRS and CTRS groups had higher response rates (63% and 57%) to 12-week PPI therapy than those with a negative composite pH (32%), indicating the predictive value of the composite pH parameter. However, we found that the number of candidate PAR episodes in the ILPRS group was significantly much lower than that in the CTRS group, and did not differ between triple pH sensors and HMII-pH catheters, suggesting a reflexogenic mechanism for symptoms generation in the ILPRS group. We also found a lower rate for the positive esophageal acid perfusion test in the ILPRS group, further supporting the distinct phenotype by the absence of esophageal symptoms and esophageal hyposensitivity to acid (Figure 3). Further studies are needed to explore the complex pathway involved in LPR symptom generation. 

### 4.3. Future: The Role of Baseline Impedance in Diagnosing Pathological Reflux

#### 4.3.1. Baseline Impedance as an Alternative in Diagnosing Pathological Reflux

Although the diagnostic role of HMII-pH and HMII-pH-based biomarkers that are relevant to treatment outcome is promising, more data including controlled trials are awaited. Recently, both ACG clinical guidelines for clinical use of esophageal physiological testing [56] and ACG clinical guidelines for the diagnosis and management of gastroesophageal reflux disease recommended impedance-pH reflux monitoring in the diagnosis of LPR [57]. Up-front ambulatory reflux monitoring of acid suppression was suggested instead of an empirical trial of PPI therapy by both guidelines in patients with ILPRS. The policy will create a considerable need for testing, as the majority of LPR patients do not have CTRS [1]. Given concerns related to expense, availability, invasiveness, and inconvenience arising from HMII-pH testing, it may not be feasible for widespread use in the future. In this regard, baseline impedance measurements are a potential alternative that measures mucosal integrity and reflects chronic reflux burden; the magnitude was inversely correlated with acid exposure time in patients with non-erosive reflux disease [58]. It could be measured through endoscopy, which has a promising future as a complimentary approach for the measurement of acid exposure time by reflux monitoring [59].

#### 4.3.2. Potential Role of Baseline Impedance in Diagnosing LPR

Distal mean nocturnal baseline impedance (MNBI) measurements from MII-pH were shown to increase the accuracy of the diagnosis of GERD compared to pH-only data as well as to predict symptomatic outcomes after PPI therapy [60,61]. It is important to explore whether MNBI in either the distal esophagus or the proximal esophagus, or even the hypopharynx may be of diagnostic value in patients with suspected LPR. Some authors found that distal, but not proximal, MNBI is significantly lower in those with evidence of acid reflux than in those without [62,63,64]; however, others showed that patients with CTRS had lower proximal MNBI when compared to those with GERD alone [65,66]. Although the role of MNBI in patients with LPR remains unclear, our recent data support that distal MNBI is lower in patients with pathological reflux, which is defined as the aforementioned composite pH. Moreover, we also found that distal MNBI is able to predict pathological reflux in both patients with CTRS and those with ILPRS [67]. From a clinical point of view, the utility of MNBI in the latter group is more relevant since they do not have esophageal symptoms and the pre-testing probability of pathological reflux is low; however, more data linked to treatment outcomes are needed.

## 5. Discussion

A literature search about the instrumental diagnosis was conducted for the prediction of treatment outcome. The selective criteria include: 1. baseline objective testing, 2. definition of predictors, 3. definition of responders at endpoint, 4. defining treatment modalities and durations, and 5. statistical significance of outcome. Of 80 identified studies, 23 met the criteria for analysis, including 1909 participants. Table 1 shows dual or triple or single pH-sensor [40,68,69,70,71,72,73], oropharyngeal pH [47,74], HMII-pH [10,31,75], MII-pH [64,65,76,77,78], salivary/ laryngeal mucosal pepsin [45,79,80], laryngoscopy [30,81], and esophagogastroduodenoscopy (EGD) [82], used in 7, 2, 3, 5, 3, 2, and 1 studies, respectively. The definitions of predictors and responders varied across studies. Among 15 studies showing significantly predictive of responders, 7 used HMII-pH or MII-pH parameters, including distal esophageal acid exposure time %, MNBI, PAR episodes, total reflux number. All three pepsin studies also showed predictive of responders. Among eight studies using HMII-pH or MII-pH, only one which consisted of 24 LPR patients using baseline PAR episodes alone failed to predict treatment response [75]. These findings corroborate the promising role of HMII-pH or MII-pH parameters and potential role of salivary pepsin test in prediction of responders to anti-reflux therapy. Thus, we proposed a management protocol for LPR based on two current ACG guidelines [56,57], i.e., the adoption of the up-front impedance-pH testing prior to anti-reflux therapy in patients with ILPRS and reserving empirical PPIs therapy in those with CTRS (Figure 4). In this protocol, we recommend EGD as the first line testing to exclude malignancy before the reflux testing, because LPR symptoms may better predict esophageal adenocarcinoma than typical reflux symptoms [83]. In addition, the findings of reflux esophagitis Los angles classification B, C, or D, peptic esophageal stricture, or Barrett’s esophagus may justify the usage of anti-reflux therapy.

## 6. Conclusions

LPR is akin to the eternally rolling boulder of King Sisyphus, and it continues to confuse patients and frustrate physicians across the fields of otolaryngology, gastroenterology, and general practice, as it has in the last three decades [11]. Unless we adopt a clinically valid diagnostic tool, as well as understand the underlying pathophysiology, the management of patients with LPR may still be very difficult given the presentation of “atypical” symptoms. The advancement of HMII-pH technology may play a promising role in precision diagnosis and make the understanding of the pathophysiology of LPR and its phenotypes possible. Other tests measuring extra-esophageal refluxate such as pepsin over airway may also be important [84,85], in concert with HMII-pH, to gather the evidence of direct airway damage. Moreover, the possibility of overlapping reflux symptoms with non-reflux etiologies cannot be underestimated; thus, objective testing is important in evaluating the necessity of long-term PPI use for refluxers and to add a therapeutic strategy for non-reflux causes. In the future, the measurement of mucosal impedance is welcome through the use of endoscopy in order to obtain evidence of pathological reflux in any patients with suspected LPR regardless of the presence or absence of typical reflux symptoms.

## Figures and Tables

**Figure 1 diagnostics-13-01643-f001:**
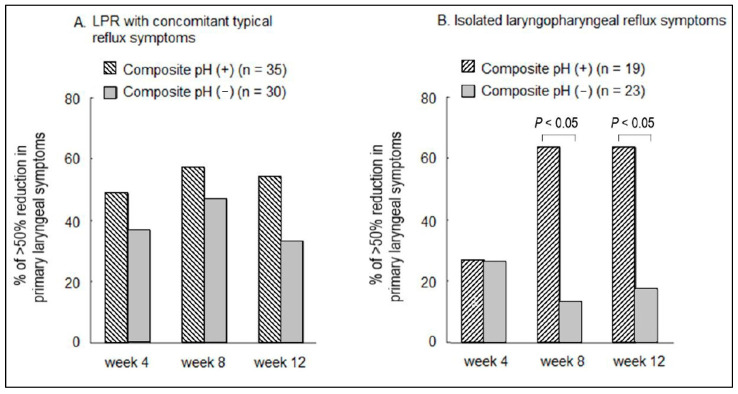
(**A**) In patients with suspected LPR and concomitant typical reflux symptoms, the pre-testing probability of a positive pH is high; thus, a positive composite pH may not predict laryngeal symptom response to PPI therapy. It is likely that factors other than reflux may also contribute to the laryngeal symptoms. (**B**) In patients with suspected isolated LPR symptoms, the pre-testing probability of a positive composite pH is low. Thus, a positive composite pH may predict laryngeal symptom response to PPI therapy and acid is likely the cause of the laryngeal symptoms [40]. LPR, laryngopharyngeal reflux; PPI, proton pump inhibitors.

**Figure 2 diagnostics-13-01643-f002:**
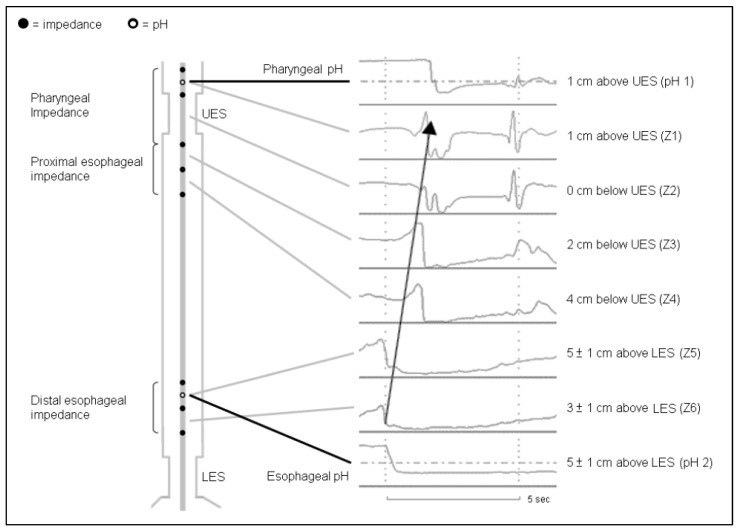
An example of pharyngeal acid reflux episodes detected by 24 h ambulatory hypopharyngeal multichannel intraluminal impedance-pH test. The mixed gas–liquid refluxate can be tracked from the distal esophagus along the entire esophagus to the hypopharynx [54]. The arrow indicates retrograde changes of pH and impedence levels.

**Figure 3 diagnostics-13-01643-f003:**
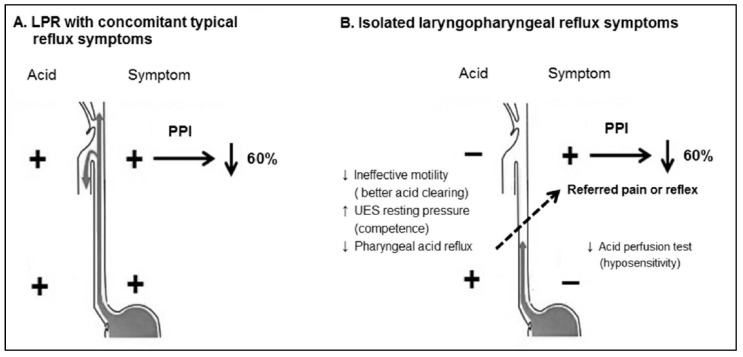
Compared to LPR patients with concomitant typical reflux symptoms (**A**), patients with isolated LPR symptoms (**B**) had fewer pharyngeal acid reflux episodes and a lower sensory response to the acid perfusion test in the distal esophagus while showing a similar symptom response rate to PPI therapy, suggesting a reflexogenic mechanism for symptoms generation [10]. The downward solid-line arrow means decrease; the oblique dotted-line arrow means vago-vagal reflex or referred pain. LPR, laryngopharyngeal reflux; PPI, proton pump inhibitors.

**Figure 4 diagnostics-13-01643-f004:**
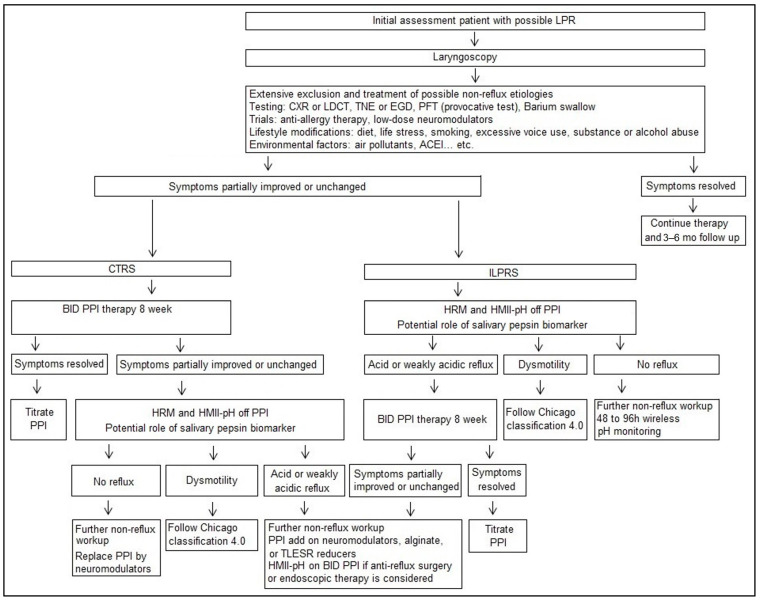
Management protocol of personalized approach for suspected LPR. LPR, laryngopharyngeal reflux; CXR, chest X-ray; LDCT, low-dose computed tomography of lungs; TNE, transnasal esophagoscopy; EGD, esophagogastroduodenoscopy; PFT, pulmonary function test; ACEI, angiotensin converting enzyme inhibitors; CTRS, concomitant typical reflux symptoms; ILPRS, isolated LPR symptoms; HRM, high resolution esophageal manometry; HMII-pH, hypopharyngeal multichannel intraluminal impedance-pH; PPI, proton pump inhibitors.

**Table 1 diagnostics-13-01643-t001:** Overviews of predictors for the treatment outcome of laryngopharyngeal reflux.

First Authors	Study Design	Case Number	Pre-Testing	Predictors	Responder Definition	Treatment Modalities/Follow-Up	Outcome
Garrigues [68]	Prospective cohort	73	Dual pH	Proximal and distal esophageal AET%	Cured laryngeal lesions and laryngeal symptoms improvement ≥ 50%	BID PPI 3 months	Non-significant
Williams [69]	Prospective cohort	20	Dual pH	1. PAR events ≥ 1; 2. distal esophageal AET > 4.9%	One level improvement of an investigator designed 4-point laryngitis grading	TID PPI 3 months	Non-significant
Vaezi [70]	Randomized controlled trial	145	Triple pH	PAR events ≥ 1	Primary symptom resolution	BID PPI 16 weeks	Non-significant
Wo [71]	Randomized controlled trial	39	Dual pH, laryngoscopy	PAR events ≥ 3; RFS	Global symptom relief	QD PPI 12 weeks	Non-significant
Qua [72]	Prospective cohort	32	Single pH, EGD, sympton alone	Erosive esophagitis, and/or, distal esophageal AET > 4.6%, and/or symptom alone	Moderate-marked laryngeal symptom improvement based on investigator-designed 4-point likert scale	BID PPI 8 weeks	67% vs. 18%, *p* = 0.026
Masaany [73]	Prospective cohort	47	Dual pH	PAR events ≥ 1	RSI imporvement ≥ 10 points or RFS improvement ≥ 5 points	BID PPI 4 months	Non-significant
Lien [40]	Prospective cohort	107	Triple pH	Presence of PAR and/or execssive esophageal acid exposure	Primary laryngeal symptoms improvement 50%	BID PPI 12 weeks	ILPRS: OR 7.9 [95% CI: 1.4–44.8]
Vailati [74]	Prospective cohort	22	Oropharyngeal pH	Ryan score > 9.4 (upright) and/or > 6.8 (supine)	RSI reduction ≥ 5 points	BID PPI 3 months	40.9% vs. 18.2%, *p* = 0.002
Yadlapati [47]	Prospective cohort	34	Oropharyngeal pH	Oropharyngeal acid exposure (below pH of 4.0, 5.0, 5.5, 6.0 and RYAN scores)	Post-treatment RSI < 13 and change in RSI ≥ 50%	QD PPI 8–12 weeks	Non-significant
Wang [31]	Prospective cohort	92	HMII-pH	1. Presence of pharyngeal bolus exposure time > 0.002%; 2. distal esophageal AET > 4%	Primary laryngeal symptoms improvement 50%	BID PPI 3 months	AET (HR: 2.55; [95%CI: 1.24–5.24]; pharyngeal bolus exposure time (HR: 2.61; [1.36–5.00])
Dulery [75]	Prospective cohort	24	HMII-pH	Pharyngeal reflux episodes ≥ 1	Primary laryngeal symptoms improvement 50%	BID PPI 8 weeks	Non-significant
Lien [10]	Prospective cohort	238	HMII-pH/triple pH	PAR events ≥ 2 and/or execssive esophageal acid exposure	Primary laryngeal symptoms improvement 50%	BID PPI 12 weeks	ILPRS: OR 4.9 [95% CI: 1.8–13.3]; CTRS: OR 4.0 [1.7–9.3]
Nennstiel [76]	Retrospecitve cohort	45	MII-pH	Distal esophageal AET > 4%, and/or total reflux number > 73	Symptom reduction ≥ 3 points of the investigator designed 10-point likert scale	BID PPI > 12 weeks	66.7% vs. 16.7% (*p* < 0.001)
Ribolsi [64]	Retrospecitve cohort	239	MII-pH	PSPW index < 61%, distal MNBI < 2292Ω	Symptom improvement >50%	BID PPI > 8 weeks	PSPW index: RR 2.4 [95% CI: 1.7–3.6]; MNBI: RR 1.9 [1.4–2.7]
Chen [65]	Retrospective cohort	63	MII-pH	Proximal and distal MNBI	Global symptom score improvement ≥ 50%	BID PPI 12 weeks	Proximal and distal MNBI (*p* < 0.001 for both)
Ribolsi [77]	Retrospecitve cohort	178	MII-pH	Erosive esophagitis, distal esophageal AET > 6%, MNBI, PSPW, typical symptoms, hypomotility, hiatal hernia	Fisman Severity Score ≤ 1	BID PPI ≥ 8 weeks	OR [95% CI]: erosive esophagitis: 3.56 [1.54–5.12], AET > 6%: 3.61 [1.42–7.63], MNBI: 3.75 [1.61–8.74), PSPW: 4.81 [2.14–10.77], typical symptoms: 1.21 [1.04–3.87], hypomotility: 3.82 [1.21–12.03], hiatal hernia: 3.48 [1.31–9.32]
Kim [78]	Prospective cohort	80	MII-pH	Proximal all reflux time and proximal longest reflux time	RSI decrease ≥ 50%	BID PPI 8 weeks	Proximal all reflux time (*p* = 0.004) and proximal longest reflux time (*p* = 0.02)
Wang [45]	Prospective cohort	74	Peptest	Peptest strong positive	RSI reduction ≥ 50%	QD PPI 8 weeks	79% vs. 50%, *p* = 0.03
Yadlapati [79]	Prospective cohort	31	Peptest	Salivary pepsin concentration	RSI ≤ 13 and/or RSI reduction > 50%	Phase 1: BID PPI 4 weeks; Phase 2: Device (reflux band) + PPI 4 weeks	High salivary pepsin concentration (*p* = 0.01)
Liu [80]	Prospective cohort	60	Interarytenoid mucosa pepsin	Moderately or strongly positive for pepsin	RSI improvement ≥ 50%	BID PPI 12 weeks	72.0% vs. 14.3% *p* < 0.01
Park [30]	Prospective cohort	85	Laryngoscopy	Pretherapy interarytenoid mucosa and true vocal folds abnormalities	Primary symptom improvement > 50%	BID PPI 4 months	Pretherapy interarytenoid mucosa and true vocal folds abnormalities (OR 1.99 [95%CI: 1.13–3.51] and 1.96 [1.13–3.39], respectivelly).
Agrawal [81]	Prospective cohort	33	Laryngoscopy	RFS and extralaryngeal score	RSI improvement ≥ 50%	QD PPI 8–12 weeks	Non-significant
Lechien [82]	Prospective cohort	148	EGD	Hiatal hernia, LES insufficiency by endoscopy	RSS reduction ≥ 20%	Various combinations, including diet, behavioral changes, PPIs, alginate, or magaldrate	Non-hiatal hernia (*p* = 0.03), LES competence (*p* = 0.03)

HMII-pH, hypopharyngeal multichannel intraluminal impedance-pH; MII-pH, multichannel intraluminal impedance-pH; EGD, esophagogastroduodenoscopy; AET, acid exposure time; PAR, pharyngeal acid reflux; MNBI, mean nocturnal baseline impedance; PSPW, post-reflux swallow-induced peristaltic wave; LES, lower esophageal sphincter; RSI, Reflux Symptom Index; RFS, Reflux Finding Score; RSS, Reflux Symptom Score; PPI, proton pump inhibitors; ILPRS, isolated laryngopharyngeal reflux symptoms; CTRS, concomitant typical reflux symptoms; OR, odds ratio; HR, hazard ratio; RR, relative risk; CI, confidence intervals.

## Data Availability

Not applicable.

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
