# Peer review of "Diagnosis of Laryngopharyngeal Reflux: Past, Present, and Future—A Mini-Review"

_diagnostics, 2023, doi:10.3390/diagnostics13091643_

Round 1

Reviewer 1 Report

A nice and technical review on the instrumental diagnosis of LPR, the text is well written and the organization of the manuscript is fine. However I would suggest to add two paragraphs on the 1) clinical presentation; 2) the role of in-office diagnosis of LPR with a detailed discussion on fiberoscopic findings, their sensitivity and specificity, the role of PPI-challenge ex adjuvantibus, and please provide a flow chart where the diagnostic pathways (possibly differentiated by the setting: for example, high-resources countries VS. less developed countries) of LPR are presented. Last please improve the discussion on the healthcare-associated diagnostic and therapeutic costs (for a lifelong treatment etc.) of LPR, which are very relevant nowadays.

Author Response

Thank you for your excellent reviewing work and precious comments.

We have written an introduction (Page 5) paragraph to briefly describe the clinical presentations, current dilemma of LPR, and the aim of this review article accordingly. We also have performed a literature review regarding the predictors of anti-reflux therapy. It is not surprising that there is currently no convincing data to support the predictive role of the larygoscopic signs for treatment outcome. Please refer to Table 1 (Page 32), as follows: anti-reflux therapy in patients with isolated laryngopharyngeal reflux symptoms.  

  1. Introduction

Laryngopharyngeal reflux (LPR) is characterized by individuals who present with chronic laryngopharyngeal symptoms such as hoarseness, vocal fatigue, excessive throat clearing, globus pharyngeus, cough, postnasal drip as well as laryngoscopic signs such as erythema, edema, ventricular obliteration, postcricoid hyperplasia, and pseudosulcus change [1]. Patients may or may not have typical reflux symptoms, therefore, may visit an otolaryngologist or a gastroenterologist, presumably depending on their primary symptoms. Various non-reflux etiologies such as voice overuse, infection, allergy, or exposure to environmental irritants may also contribute to the similar symptoms and signs. Despite the development of “disease-specific” instruments to measure the disease severity such as the Reflux Symptom Index (RSI) [2] and the Reflux Finding Score [3], the symptoms and signs remain “non-specific”. As a result, reflux itself is just one of a myriad of causes which irritate the lining of aerodigestive tract. LPR is a prevalent disease which was estimated to be 10% of the outpatients in the otolaryngology units [1]. The quality of life of LPR patients is generally poor [4], however, the management is challenging. Traditionally, empirical proton pump inhibitors (PPIs) once or twice daily is often a pragmatic therapeutic strategy and those who are refractory to high dose PPIs treatment are recommended to refer for the reflux testing [5]. Such algorithm has recently been challenged by the up-front testing using impedance-pH and manometry prior to anti-reflux therapy in order to minimize the cost [6]. Moreover, there are discrepancies between otolaryngology and gastroenterology guidelines regarding the indications of acid suppression therapy [1,7]. The gastroenterology guidelines recommend against acid suppression therapy in patients with isolated LPR symptoms because there is scarce evidence to show the superiority of PPIs to placebo in controlled trials, while the otolaryngology guideline states that the majority of LPR patients do not have heartburn or esophagitis, i.e., isolated LPR symptoms. Recent Lyon consensus for diagnosis of gastroesophageal reflux disease (GERD) also question the utility of proximal esophageal or pharyngeal testing because of the lack of consistent outcome studies [8]. The aim of this review is to discuss the evolution of objective testing for LPR and its predictive role on anti-reflux therapy.

Table 1. Overviews of predictors for the treatment outcome of laryngopharyngeal reflux

First authors

Study design

Case

number

Pre-testing

Predictors

Responder definition

Treatment modalities/follow-up

Outcome

Garrigues [68[

Prospective cohort

73

Dual pH

Proximal & distal esophageal AET%

Cured laryngeal lesions and laryngeal symptoms improvement ≥ 50%

BID PPI 3 months

Non-significant

Williams [69]

Prospective cohort

20

Dual pH

1. PAR events ≥ 1 ;

2. distal esophageal AET > 4.9%

One level improvement of an investigator designed 4-point laryngitis grading

TID PPI 3 months

Non-significant

Vaezi [70]

Randomized controlled trial

145

Triple pH

PAR events ≥ 1

Primary symptom resolution

BID PPI 16 weeks

Non-significant

Wo [71]

Randomized controlled trial

39

Dual pH, laryngoscopy

PAR events ≥ 3; RFS

Global symptom relief

QD PPI 12 weeks

Non-significant

Qua [72]

Prospective cohort

32

Single pH, EGD, sympton alone

Erosive esophagitis, and/or, distal esophageal AET > 4.6%, and/or symptom alone

Moderate-marked laryngeal symptom improvement based on investigator-designed 4-point likert scale

BID PPI 8 weeks

67% vs 18%, P = 0.026

Masaany [73]

Prospective cohort

47

Dual pH

PAR events ≥ 1

RSI imporvement ≥ 10 points or RFS improvement ≥ 5 points

BID PPI 4 months

Non-significant

Lien [40]

Prospective cohort

107

Triple pH

Presence of PAR and/or execssive esophageal acid exposure

Primary laryngeal symptoms improvement 50%

BID PPI 12 weeks

ILPRS: OR 7.9 [95% CI: 1.4–44.8]

Vailati [74]

Prospective cohort

22

Oropharyngeal pH

Ryan score > 9.4 (upright) and/or > 6.8 (supine)

RSI reduction ≥ 5 points

BID PPI 3 months

40.9% vs 18.2%, P = 0.002

Yadlapati [47]

Prospective cohort

34

Oropharyngeal pH

Oropharyngeal acid exposure (below pH of 4.0, 5.0, 5.5, 6.0 and RYAN scores)

Post-treatment RSI < 13 and change in RSI ≥ 50%

QD PPI 8-12 weeks

Non-significant

Wang [31]

Prospective cohort

92

HMII-pH

1. Presence of pharyngeal bolus exposure time > 0.002% ;

2. distal esophageal AET > 4%

Primary laryngeal symptoms improvement 50%

BID PPI 3 months

AET (HR: 2.55; [95%CI: 1.24–5.24]; pharyngeal bolus exposure time (HR: 2.61; [1.36–5.00])

Dulery [75]

Prospective cohort

24

HMII-pH

Pharyngeal reflux episodes ≥ 1

Primary laryngeal symptoms improvement 50%

BID PPI 8 weeks

Non-significant

Lien [10]

Prospective cohort

238

HMII-pH/triple pH

PAR events ≥ 2 and/or execssive esophageal acid exposure

Primary laryngeal symptoms improvement 50%

BID PPI 12 weeks

ILPRS: OR 4.9 [95% CI: 1.8-13.3]; CTRS: OR 4.0 [1.7-9.3]

Nennstiel [76]

Retrospecitve cohort

45

MII-pH

Distal esophageal AET > 4%, and/or total reflux number > 73

Symptom reduction ≥ 3 points of the investigator designed 10-point likert scale

BID PPI > 12 weeks

66.7% vs 16.7% (P < 0.001)

Ribolsi [64]

Retrospecitve cohort

239

MII-pH

PSPW index < 61%, distal MNBI < 2292Ω

Symptom improvement >50%

BID PPI > 8 weeks

PSPW index: RR 2.4 [95% CI: 1.7–3.6]; MNBI: RR 1.9 [1.4–2.7]

Chen [65]

Retrospective cohort

63

MII-pH

Proximal and distal MNBI

Global symptom score improvement ≥ 50%

BID PPI 12 weeks

Proximal and distal MNBI (P < 0.001 for both)

Ribolsi [77]

Retrospecitve cohort

178

MII-pH

Erosive esophagitis, distal esophageal AET > 6%, MNBI, PSPW, typical symptoms, hypomotility, hiatal hernia

Fisman Severity Score ≤ 1

BID PPI ≥ 8 weeks

OR [95% CI]: erosive esophagitis: 3.56 [1.54–5.12], AET > 6%: 3.61 [1.42–7.63], MNBI: 3.75 [1.61–8.74), PSPW: 4.81 [2.14–10.77], typical symptoms: 1.21 [1.04–3.87],hypomotility: 3.82 [1.21–12.03], hiatal hernia: 3.48 [1.31–9.32]

Kim [78]

Prospective cohort

80

MII-pH

Proximal all reflux time and proximal longest reflux time

RSI decrease ≥ 50%

BID PPI 8 weeks

Proximal all reflux time (P = 0.004) and proximal longest reflux time (P = 0.02)

Wang [45]

Prospective cohort

74

Peptest

Peptest strong positive

RSI reduction ≥ 50%

QD PPI 8 weeks

79% vs 50%, P = 0.03

Yadlapati [79]

Prospective cohort

31

Peptest

Salivary pepsin concentration

RSI ≤ 13 and/or RSI reduction > 50%

Phase 1: BID PPI 4 weeks; Phase 2: Device (reflux band) + PPI 4 weeks

High salivary pepsin concentration (P = 0.01)

Liu [80]

Prospective cohort

60

Interarytenoid mucosa pepsin

Moderately or strongly positive for pepsin

RSI improvement ≥ 50%

BID PPI 12 weeks

72.0% vs 14.3% P < 0.01

Park [30]

Prospective cohort

85

Laryngoscopy

Pretherapy interarytenoid mucosa and true vocal folds abnormalities

Primary symptom improvement > 50%

BID PPI 4 months

Pretherapy interarytenoid mucosa and true vocal folds abnormalities (OR 1.99 [95%CI: 1.13-3.51] and 1.96 [1.13-3.39], respectivelly).

Agrawal [81]

Prospective cohort

33

Laryngoscopy

RFS and extralaryngeal score

RSI improvement ≥ 50%

QD PPI 8-12 weeks

Non-significant

Lechien [82]

Prospective cohort

148

EGD

Hiatal hernia, LES insufficiency by endoscopy

RSS reduction ≥ 20%

Various combinations, including diet, behavioral changes,PPIs, alginate, or magaldrate

Non-hiatal hernia (P = 0.03), LES competence (P = 0.03)

HMII-pH, hypopharyngeal multichannel intraluminal impedance-pH; MII-pH, multichannel intraluminal impedance-pH; EGD, esophagogastroduodenoscopy; AET, acid exposure time; PAR, pharyngeal acid reflux; MNBI, mean nocturnal baseline impedance; PSPW, post-reflux swallow-induced peristaltic wave; LES, lower esophageal sphincter; RSI, Reflux Symptom Index; RFS, Reflux Finding Score; RSS, Reflux Symptom Score; PPI, proton pump inhibitors; ILPRS, isolated laryngopharyngeal reflux symptoms; CTRS, concomitant typical reflux symptoms; OR, odds ratio; HR, hazard ratio; RR, relative risk; CI, confidence intervals.

We have proposed a management protocol for personalized approach of suspected LPR in Figure 4 (Page 39). However, this protocol may only be suitable for highly developed countries currently. More research are needed for those low developed counties in the future when the pathophysiology is better understood. In addition, we have also written a discussion (Page 18) paragraph to explain Table 1 and Figure 4, as follows:

Figure 4. Management protocol of personalized approach for suspected LPR. LPR, laryngopharyngeal reflux; CXR, chest X-ray; LDCT, low-dose computed tomography of lungs; TNE, transnasal esophagoscopy; EGD, esophagogastroduodenoscopy; PFT, pulmonary function test; ACEI, angiotensin converting enzyme inhibitors; CTRS, concomitant typical reflux symptoms; ILPRS, isolated LPR symptoms; HRM, high resolution esophageal manometry; HMII-pH, hypopharyngeal multichannel intraluminal impedance-pH; PPI, proton pump inhibitors.

  1. Discussion

    A literature search about the instrumental diagnosis was conducted for the prediction of treatment outcome. The selective criteria include: 1. baseline objective testing, 2. definition of predictors, 3. definition of responders at endpoint 4. defining treatment modalities and durations, 5. statistical significance of outcome. Of 80 identified studies, 23 met criteria for analysis, including 1,909 subjects. Table 1 shows dual or triple or single pH-sensor [40,68-73], oropharyngeal pH [47,74], HMII-pH [10,31,75], MII-pH [64,65,76-78], salivary/ laryngeal mucosal pepsin [45,79,80], laryngoscopy [30,81], and esophagogastroduodenoscopy (EGD) [82], used in 7, 2, 3, 5, 3, 2, and 1 studies, respectively. The definitions of predictors and responders vary across studies. Among 15 studies showing significantly predictive of responders, 7 used HMII-pH or MII-pH parameters, including distal esophageal acid exposure time %, MNBI, PAR episodes, total reflux number. All 3 pepsin studies also show predictive of responders. Among 8 studies using HMII-pH or MII-pH, only one which consisted 24 LPR patients using baseline PAR episodes alone failed to predict treatment response [75]. These findings corroborate the promising role of HMII-pH or MII-pH parameters and potential role of salivary pepsin test in prediction of responders to anti-reflux therapy. Thus, we proposed a management protocol for LPR based on two current ACG guidelines [56,57], i.e. adoption of the up-front impedance-pH testing prior to anti-reflux therapy in patients with ILPRS and reserving empirical PPIs therapy in those with CTRS [Figure 4]. In this protocol, we recommend EGD as the first line testing to exclude malignancy before the reflux testing, because LPR symptoms may better predict esophageal adenocarcinoma than typical reflux symptoms [83]. In addition, the findings of reflux esophagitis Los angles classification B, C, or D, peptic esophageal stricture, or Barrett’s esophagus may justify the usage of anti-reflux therapy.

Reviewer 2 Report

The authors deal with an important topic of otorhinolaryngology, concerning the objective diagnosis of laryngopharyngeal reflux (LPR). The familiarity of the authors with this theme is clear, also evident from their cited works. In any case, the review seems to lack some features that could increase its scientific relevance.

-First of all, the review is not a systematic review; the authors did not clarify the inclusion and exclusion criteria of the studies mentioned relating to the topic; in some points it even seems that this is a review of their previous studies rather than a review of the literature. In this regard, I recommend reviewing the existing clinical studies for each diagnostic method, defining the selection criteria of the studies and therefore drawing conclusions that are based on evidence (PRISMA method). In any case, I think that it is the editor's judgment, on the basis of the characteristics of the journal, whether to accept a well-written but not systematic review. 

-The aim of the review is not clear and should be better specified by the authors both in the abstract and in the text. In some cases, they mentioned studies relating to the specificity of the diagnostic methods for reflux, in other cases, studies on the predictability of the same diagnostic methods of the efficacy of the anti-reflux therapy. I believe that studies with different objectives should be more carefully separated in the review for each diagnostic method examined.

-The authors never mentioned the endoscopy as a diagnostic method despite LPR has characteristic signs well described with the reflux finding score (RFS). They should therefore review the studies in the literature relating to this diagnostic method, to the predictivity of endoscopic signs of the responsiveness to therapy and therefore its comparison with other diagnostic methods.

I believe, however, that a clear result in defining the best diagnostic method could be achieved only with a more careful and systematic review of the literature.

Author Response

Reviewer 2:

The authors deal with an important topic of otorhinolaryngology, concerning the objective diagnosis of laryngopharyngeal reflux (LPR). The familiarity of the authors with this theme is clear, also evident from their cited works. In any case, the review seems to lack some features that could increase its scientific relevance.

  1. First of all, the review is not a systematic review; the authors did not clarify the inclusion and exclusion criteria of the studies mentioned relating to the topic; in some points it even seems that this is a review of their previous studies rather than a review of the literature. In this regard, I recommend reviewing the existing clinical studies for each diagnostic method, defining the selection criteria of the studies and therefore drawing conclusions that are based on evidence (PRISMA method). In any case, I think that it is the editor's judgment, on the basis of the characteristics of the journal, whether to accept a well-written but not systematic review. 
  2. The aim of the review is not clear and should be better specified by the authors both in the abstract and in the text. In some cases, they mentioned studies relating to the specificity of the diagnostic methods for reflux, in other cases, studies on the predictability of the same diagnostic methods of the efficacy of the anti-reflux therapy. I believe that studies with different objectives should be more carefully separated in the review for each diagnostic method examined.
  3. The authors never mentioned the endoscopy as a diagnostic method despite LPR has characteristic signs well described with the reflux finding score (RFS). They should therefore review the studies in the literature relating to this diagnostic method, to the predictivity of endoscopic signs of the responsiveness to therapy and therefore its comparison with other diagnostic methods.

 I believe, however, that a clear result in defining the best diagnostic method could be achieved only with a more careful and systematic review of the literature.

In reply to Reviewer 2:

  1. Thank you very much for your constructive criticisms and precious comments. We agree that a systematic review is a scientifically rigorous method, while a mini-review is likely to be subject to bias. However, the lack of a gold standard for diagnosis in LPR has resulted in over-diagnosis as well as over-treatment in this heterogeneous patient group. Our study aimed to discuss the evolution of objective diagnostic methods and their predictive values of treatment outcome. According to your recommendations, we have performed a literature review focusing on the predictors of treatment responders which is shown in Table 1 (Page 32) and the selective criteria which is presented in the discussion (Page 18) paragraph, as follows:

  1. Discussion

    A literature search about the instrumental diagnosis was conducted for the prediction of treatment outcome. The selective criteria include: 1. baseline objective testing, 2. definition of predictors, 3. definition of responders at endpoint 4. defining treatment modalities and durations, 5. statistical significance of outcome. Of 80 identified studies, 23 met criteria for analysis, including 1,909 subjects. Table 1 shows dual or triple or single pH-sensor [40,68-73], oropharyngeal pH [47,74], HMII-pH [10,31,75], MII-pH [64,65,76-78], salivary/ laryngeal mucosal pepsin [45,79,80], laryngoscopy [30,81], and esophagogastroduodenoscopy (EGD) [82], used in 7, 2, 3, 5, 3, 2, and 1 studies, respectively. The definitions of predictors and responders vary across studies. Among 15 studies showing significantly predictive of responders, 7 used HMII-pH or MII-pH parameters, including distal esophageal acid exposure time %, MNBI, PAR episodes, total reflux number. All 3 pepsin studies also show predictive of responders. Among 8 studies using HMII-pH or MII-pH, only one which consisted 24 LPR patients using baseline PAR episodes alone failed to predict treatment response [75]. These findings corroborate the promising role of HMII-pH or MII-pH parameters and potential role of salivary pepsin test in prediction of responders to anti-reflux therapy. Thus, we proposed a management protocol for LPR based on two current ACG guidelines [56,57], i.e. adoption of the up-front impedance-pH testing prior to anti-reflux therapy in patients with ILPRS and reserving empirical PPIs therapy in those with CTRS [Figure 4]. In this protocol, we recommend EGD as the first line testing to exclude malignancy before the reflux testing, because LPR symptoms may better predict esophageal adenocarcinoma than typical reflux symptoms.[83] In addition, the findings of reflux esophagitis Los angles classification B, C, or D, peptic esophageal stricture, or Barrett’s esophagus may justify the usage of anti-reflux therapy.

Table 1. Overviews of predictors for the treatment outcome of laryngopharyngeal reflux

First authors

Study design

Case

number

Pre-testing

Predictors

Responder definition

Treatment modalities/follow-up

Outcome

Garrigues [68[

Prospective cohort

73

Dual pH

Proximal & distal esophageal AET%

Cured laryngeal lesions and laryngeal symptoms improvement ≥ 50%

BID PPI 3 months

Non-significant

Williams [69]

Prospective cohort

20

Dual pH

1. PAR events ≥ 1 ;

2. distal esophageal AET > 4.9%

One level improvement of an investigator designed 4-point laryngitis grading

TID PPI 3 months

Non-significant

Vaezi [70]

Randomized controlled trial

145

Triple pH

PAR events ≥ 1

Primary symptom resolution

BID PPI 16 weeks

Non-significant

Wo [71]

Randomized controlled trial

39

Dual pH, laryngoscopy

PAR events ≥ 3; RFS

Global symptom relief

QD PPI 12 weeks

Non-significant

Qua [72]

Prospective cohort

32

Single pH, EGD, sympton alone

Erosive esophagitis, and/or, distal esophageal AET > 4.6%, and/or symptom alone

Moderate-marked laryngeal symptom improvement based on investigator-designed 4-point likert scale

BID PPI 8 weeks

67% vs 18%, P = 0.026

Masaany [73]

Prospective cohort

47

Dual pH

PAR events ≥ 1

RSI imporvement ≥ 10 points or RFS improvement ≥ 5 points

BID PPI 4 months

Non-significant

Lien [40]

Prospective cohort

107

Triple pH

Presence of PAR and/or execssive esophageal acid exposure

Primary laryngeal symptoms improvement 50%

BID PPI 12 weeks

ILPRS: OR 7.9 [95% CI: 1.4–44.8]

Vailati [74]

Prospective cohort

22

Oropharyngeal pH

Ryan score > 9.4 (upright) and/or > 6.8 (supine)

RSI reduction ≥ 5 points

BID PPI 3 months

40.9% vs 18.2%, P = 0.002

Yadlapati [47]

Prospective cohort

34

Oropharyngeal pH

Oropharyngeal acid exposure (below pH of 4.0, 5.0, 5.5, 6.0 and RYAN scores)

Post-treatment RSI < 13 and change in RSI ≥ 50%

QD PPI 8-12 weeks

Non-significant

Wang [31]

Prospective cohort

92

HMII-pH

1. Presence of pharyngeal bolus exposure time > 0.002% ;

2. distal esophageal AET > 4%

Primary laryngeal symptoms improvement 50%

BID PPI 3 months

AET (HR: 2.55; [95%CI: 1.24–5.24]; pharyngeal bolus exposure time (HR: 2.61; [1.36–5.00])

Dulery [75]

Prospective cohort

24

HMII-pH

Pharyngeal reflux episodes ≥ 1

Primary laryngeal symptoms improvement 50%

BID PPI 8 weeks

Non-significant

Lien [10]

Prospective cohort

238

HMII-pH/triple pH

PAR events ≥ 2 and/or execssive esophageal acid exposure

Primary laryngeal symptoms improvement 50%

BID PPI 12 weeks

ILPRS: OR 4.9 [95% CI: 1.8-13.3]; CTRS: OR 4.0 [1.7-9.3]

Nennstiel [76]

Retrospecitve cohort

45

MII-pH

Distal esophageal AET > 4%, and/or total reflux number > 73

Symptom reduction ≥ 3 points of the investigator designed 10-point likert scale

BID PPI > 12 weeks

66.7% vs 16.7% (P < 0.001)

Ribolsi [64]

Retrospecitve cohort

239

MII-pH

PSPW index < 61%, distal MNBI < 2292Ω

Symptom improvement >50%

BID PPI > 8 weeks

PSPW index: RR 2.4 [95% CI: 1.7–3.6]; MNBI: RR 1.9 [1.4–2.7]

Chen [65]

Retrospective cohort

63

MII-pH

Proximal and distal MNBI

Global symptom score improvement ≥ 50%

BID PPI 12 weeks

Proximal and distal MNBI (P < 0.001 for both)

Ribolsi [77]

Retrospecitve cohort

178

MII-pH

Erosive esophagitis, distal esophageal AET > 6%, MNBI, PSPW, typical symptoms, hypomotility, hiatal hernia

Fisman Severity Score ≤ 1

BID PPI ≥ 8 weeks

OR [95% CI]: erosive esophagitis: 3.56 [1.54–5.12], AET > 6%: 3.61 [1.42–7.63], MNBI: 3.75 [1.61–8.74), PSPW: 4.81 [2.14–10.77], typical symptoms: 1.21 [1.04–3.87],hypomotility: 3.82 [1.21–12.03], hiatal hernia: 3.48 [1.31–9.32]

Kim [78]

Prospective cohort

80

MII-pH

Proximal all reflux time and proximal longest reflux time

RSI decrease ≥ 50%

BID PPI 8 weeks

Proximal all reflux time (P = 0.004) and proximal longest reflux time (P = 0.02)

Wang [45]

Prospective cohort

74

Peptest

Peptest strong positive

RSI reduction ≥ 50%

QD PPI 8 weeks

79% vs 50%, P = 0.03

Yadlapati [79]

Prospective cohort

31

Peptest

Salivary pepsin concentration

RSI ≤ 13 and/or RSI reduction > 50%

Phase 1: BID PPI 4 weeks; Phase 2: Device (reflux band) + PPI 4 weeks

High salivary pepsin concentration (P = 0.01)

Liu [80]

Prospective cohort

60

Interarytenoid mucosa pepsin

Moderately or strongly positive for pepsin

RSI improvement ≥ 50%

BID PPI 12 weeks

72.0% vs 14.3% P < 0.01

Park [30]

Prospective cohort

85

Laryngoscopy

Pretherapy interarytenoid mucosa and true vocal folds abnormalities

Primary symptom improvement > 50%

BID PPI 4 months

Pretherapy interarytenoid mucosa and true vocal folds abnormalities (OR 1.99 [95%CI: 1.13-3.51] and 1.96 [1.13-3.39], respectivelly).

Agrawal [81]

Prospective cohort

33

Laryngoscopy

RFS and extralaryngeal score

RSI improvement ≥ 50%

QD PPI 8-12 weeks

Non-significant

Lechien [82]

Prospective cohort

148

EGD

Hiatal hernia, LES insufficiency by endoscopy

RSS reduction ≥ 20%

Various combinations, including diet, behavioral changes,PPIs, alginate, or magaldrate

Non-hiatal hernia (P = 0.03), LES competence (P = 0.03)

HMII-pH, hypopharyngeal multichannel intraluminal impedance-pH; MII-pH, multichannel intraluminal impedance-pH; EGD, esophagogastroduodenoscopy; AET, acid exposure time; PAR, pharyngeal acid reflux; MNBI, mean nocturnal baseline impedance; PSPW, post-reflux swallow-induced peristaltic wave; LES, lower esophageal sphincter; RSI, Reflux Symptom Index; RFS, Reflux Finding Score; RSS, Reflux Symptom Score; PPI, proton pump inhibitors; ILPRS, isolated laryngopharyngeal reflux symptoms; CTRS, concomitant typical reflux symptoms; OR, odds ratio; HR, hazard ratio; RR, relative risk; CI, confidence intervals.

Thank you for the precious recommendations. We have added the aims of this review in the abstract (Page 3) and the introduction (Page 5) paragraph, as follows:

Abstract: Laryngopharyngeal reflux (LPR) is a variant of gastroesophageal reflux disease (GERD) in which gastric refluxate irritates the lining of the aerodigestive tract and causes troublesome airway symptoms or complications. LPR is a prevalent disease that creates a significant socioeconomic burden due to its negative impact on quality of life, tremendous medical expense, and possible cancer risk. Although treatment modalities are similar between LPR and GERD, the diagnosis of LPR is more challenging than GERD due to its non-specific symptoms/signs. Due to the lack of pathognomonic features of endoscopy, mounting evidence has focused on physiological diagnostic testing. Two decades ago, a dual pH probe was considered the gold standard for detecting pharyngeal acidic reflux episodes. Despite an association with LPR, the dual pH was unable to predict the treatment response in clinical practice, presumably due to frequently encountered artifacts. Currently, hypopharygneal multichannel intraluminal impedance-pH catheters incorporating two trans-upper esophageal sphincter impedance sensors enable to differentiate pharyngeal refluxes from swallows. The validation of pharyngeal acid reflux episodes that are relevant to anti-reflux treatment is therefore crucial. Given no diagnostic gold standard of LPR, this review article aimed to discuss the evolution of objective diagnostic testing and its predictive role of treatment response.

  1. Introduction

Laryngopharyngeal reflux (LPR) is characterized by individuals who present with chronic laryngopharyngeal symptoms such as hoarseness, vocal fatigue, excessive throat clearing, globus pharyngeus, cough, postnasal drip as well as laryngoscopic signs such as erythema, edema, ventricular obliteration, postcricoid hyperplasia, and pseudosulcus change [1]. Patients may or may not have typical reflux symptoms, therefore, may visit an otolaryngologist or a gastroenterologist, presumably depending on their primary symptoms. Various non-reflux etiologies such as voice overuse, infection, allergy, or exposure to environmental irritants may also contribute to the similar symptoms and signs. Despite the development of “disease-specific” instruments to measure the disease severity such as the Reflux Symptom Index (RSI) [2] and the Reflux Finding Score [3], the symptoms and signs remain “non-specific”. As a result, reflux itself is just one of a myriad of causes which irritate the lining of aerodigestive tract. LPR is a prevalent disease which was estimated to be 10% of the outpatients in the otolaryngology units [1]. The quality of life of LPR patients is generally poor [4], however, the management is challenging. Traditionally, empirical proton pump inhibitors (PPIs) once or twice daily is often a pragmatic therapeutic strategy and those who are refractory to high dose PPIs treatment are recommended to refer for the reflux testing [5]. Such algorithm has recently been challenged by the up-front testing using impedance-pH and manometry prior to anti-reflux therapy in order to minimize the cost [6]. Moreover, there are discrepancies between otolaryngology and gastroenterology guidelines regarding the indications of acid suppression therapy [1,7]. The gastroenterology guidelines recommend against acid suppression therapy in patients with isolated LPR symptoms because there is scarce evidence to show the superiority of PPIs to placebo in controlled trials, while the otolaryngology guideline states that the majority of LPR patients do not have heartburn or esophagitis, i.e., isolated LPR symptoms. Recent Lyon consensus for diagnosis of gastroesophageal reflux disease (GERD) also question the utility of proximal esophageal or pharyngeal testing because of the lack of consistent outcome studies [8]. The aim of this review is to discuss the evolution of objective testing for LPR and its predictive role on anti-reflux therapy.

  1. Thank you for the important question regarding the diagnostic role of laryngoscopy in patients with suspected LPR. Unlike reflux esophagitis in GERD, to the best of our knowledge, there are no specific signs for LPR with predictive value. Even the “disease-specific” instruments such as the Reflux Finding Score may probably only measure the severity of various etiologies. However, it may still be important as the first line testing to exclude malignancy.

As the role of MII-pH in diagnosing GERD stated in Lyon consensus, we hope that the HMII-pH will be a useful tool in the management of LPR.  

Again, we deeply appreciate your excellent review work and valuable recommendations.
